# GROUP PATTERN SELECTION OPTIMIZATION: LET LRMS PICK THE RIGHT PATTERN FOR REASONING

## ABSTRACT

Large reasoning models (LRMs) exhibit diverse high-level reasoning patterns (e.g., direct solution, reflection-and-verification, and exploring multiple solutions), yet prevailing training recipes implicitly bias models toward a few dominant patterns. We systematically analyze these patterns and observe substantial accuracy variance across patterns on math and science benchmarks, implying that the pattern a model adopts is often sub-optimal for a given problem. We introduce Group Pattern Selection Optimal (GPSO), a reinforcement-learning framework that extends GRPO with multi-pattern rollout, optimal pattern selection per problem via verifier signals, and attention masking to prevent leakage from explicit pattern suffixes into the learned policy. By intelligently exploring a portfolio of diverse patterns and optimizing the model's policy on the most effective ones, GPSO learns the intrinsic mapping from problem to pattern. Extensive experiments demonstrate that GPSO delivers consistent and substantial performance improvements across a wide range of model backbones and reasoning benchmarks, effectively mitigating the sub-optimality issue and enabling more robust and adaptable LLM reasoning.

## 1 INTRODUCTION

Recent advances in Large Language Models (LLMs), particularly those focused on complex reasoning, have yielded remarkable capabilities in solving challenging tasks across mathematics, science, and programming. Models like DeepSeek-R1 Zhang et al. (2023) and OpenAI-o1 OpenAI (2024) demonstrate a new paradigm of reasoning, where they generate long, multi-step Chain-of-Thought (CoT) Wei et al. (2022) responses. A critical enabling factor behind this emergent behavior is the use of reinforcement learning (RL) from human feedback or model-generated trajectories, with algorithms such as Proximal Policy Optimization (PPO) Schulman et al. (2017) and GRPO Shao et al. (2024) playing a central role. These training paradigms encourage models to explore, self-correct, and refine their reasoning on the fly, leading to impressive performance gains.

Inspired by these successes, a growing body of research has turned its attention to understanding the internal reasoning patterns adopted by these models. These reasoning patterns, or "paradigms," refer to the high-level, observable strategies a model employs to navigate a complex problem space, such as providing direct answers, decomposing problems, exploring alternative solutions, or employing tools. Several studies have systematically analyzed the cognitive behaviors of LLMs, revealing a rich spectrum of patterns such as self-reflection, backtracking, and exploration of multiple hypotheses (Wen et al., 2025b; Gandhi et al., 2025). Crucially, the reasoning patterns these models learn typically do not emerge spontaneously from scratch. Instead, they are shaped during the cold-start phase through human-designed prompts or explicitly reinforced by human preferences during reinforcement learning. For example, Chen et al. (2025b) analyzed the evolution of these patterns before and after RL fine-tuning, finding that trained models tend to converge on a limited set of high-success-rate patterns. This observation leads us to a crucial, unaddressed question: **Are the reasoning patterns chosen by LRMs truly optimal for problem solving?**

To answer this question, we conduct a comprehensive empirical study. First, we perform a systematic analysis of the reasoning trajectories generated by seven state-of-the-art LLMs across mathematics, science, and code domains. Our analysis reveals that while LLMs possess the potential for diverse reasoning, they consistently default to a limited set of dominant patterns. Specifically, we

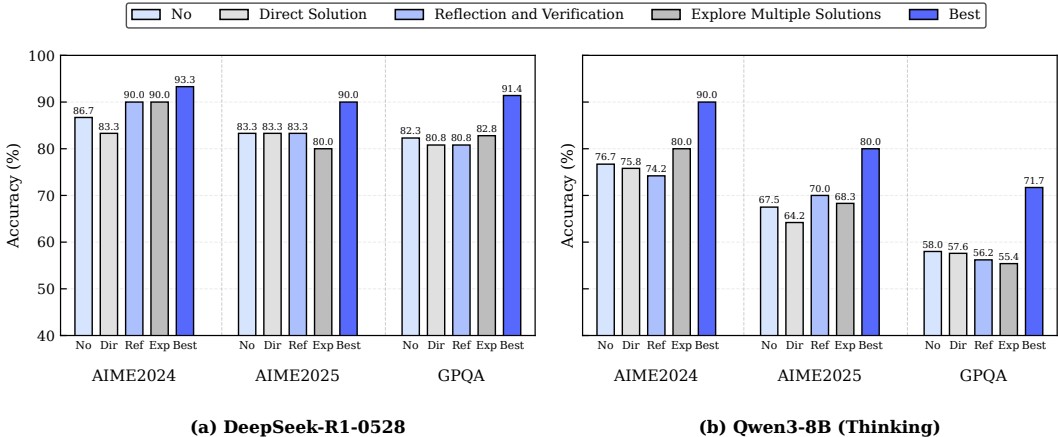

Figure 1: Comparison of model performance under different reasoning patterns on three benchmarks (AIME2024, AIME2025, and GPQA): (a) **DeepSeek-R1-0528** and (b) **Qwen3-8B (Thinking)**. **No**: No reasoning prompt, **Dir**: Direct solution, **Ref**: Reflection and verification, **Exp**: Explore Multiple solutions, **Best**: Pattern selected with the highest accuracy on each question.

find that the majority of reasoning trajectories can be classified into three high-level categories: *Direct Solution*, *Reflection and Verification*, and *Exploration of Multiple Solutions*. Interestingly, we observed that Reflection and Verification emerges as the default and primary reasoning pattern for most models, likely due to its robustness in self-correction (details can be found in Appendix A).

Subsequently, we evaluate the performance of two high-performing LRMs (DeepSeek-R1-0528 and Qwen3-8B-Thinking) under these distinct reasoning patterns. They generate solutions using each of the three patterns through tailored, in-context prompts. The results, as illustrated in Figure 1, reveal a striking finding: model performance varies significantly across different reasoning patterns. For instance, while Reflection and Verification might be optimal for some problems, Exploration of Multiple Solutions often yields substantially higher accuracy on tasks requiring novel insights. Critically, our results demonstrate that if LLMs were capable of dynamically selecting the most suitable pattern for each problem and outputting the best-performing trajectory, their overall performance could be enhanced by a substantial margin. This leads us to our core conclusion: **The reasoning patterns chosen by LRMs are not optimal.**

In this paper, we propose Group Pattern Selection Optimization (GPSO), a novel training paradigm that teaches the model to intelligently select the optimal reasoning pattern for a given problem. Our method extends GRPO by incorporating multi-pattern exploration and optimal pattern optimization. During training, GPSO dynamically evaluates multiple candidate reasoning patterns for each problem. It then identifies the most effective pattern based on verifier-based signals and updates the model policy specifically on the rollouts of this optimal pattern. To ensure that the model learns the intrinsic mapping from problem to pattern—rather than overfitting to explicit pattern tokens—GPSO employs a gradient masking technique. This mechanism ensures that the explicit prompts used as exploration scaffolds do not leak into the learned policy, allowing the model to internally select the appropriate pattern on its own during reasoning. Through extensive experiments, we demonstrate that GPSO significantly outperforms existing methods and effectively addresses the sub-optimality issue in LLM reasoning.

Experimental results demonstrate that our proposed GPSO brings consistent and substantial improvements across diverse model backbones and reasoning benchmarks. As summarized in Table 1, applying GPSO significantly boosts the performance of various LLMs. For example, it improves the average performance of Nemotron-1.5B from 55.4 to 58.0, a relative gain of +2.6%. Similarly, DeepSeek-Qwen-7B sees an increase from 55.6 to 58.7 (+3.1%), while DeepSeek-LLaMA-8B improves from 51.4 to 54.6 (+3.2%). Moreover, our method proves highly effective even on the strongest baseline model in our evaluation suite, Qwen3-8B (Thinking), which achieves its highest overall score of 75.3 after being fine-tuned with GPSO. These results strongly validate our hypothe-

sis and demonstrate that GPSO is both a model-agnostic and consistently effective training paradigm for optimizing reasoning performance.

## 2 RELATED WORK

This section reviews key research areas that form the foundation of our work: Reinforcement Learning with Verifiable Rewards (RLVR), sampling strategies for RL fine-tuning, and the study of reasoning patterns in large language models. Our study builds upon these fields by proposing a novel method to address the sub-optimal reasoning patterns that emerge from current RLVR training paradigms.

### 2.1 REINFORCEMENT LEARNING WITH VERIFIABLE REWARDS

RLVR has emerged as a powerful and scalable post-training paradigm for large language models by leveraging rule-based or executable feedback, such as program execution results or logical consistency checks (Ouyang et al., 2022; Bai et al., 2022). This approach bypasses the reliance on costly human-annotated reward models, showing strong improvements in reasoning-heavy domains like symbolic mathematics and code generation (Wang et al., 2025; Chen et al., 2025c). The success of models like DeepSeek-R1 DeepSeek-AI et al. (2025), which was trained with the GRPO algorithm Shao et al. (2024), has inspired a surge of follow-up research (He et al., 2025; Tang et al., 2025a; Cheng et al., 2025). The researchers conduct in-depth studies on the design and robustness of the reward function in RLVR Su et al. (2025); Li et al. (2025a); Zhang et al. (2025a), the efficient utilization of data Tang et al. (2025b); Yang et al. (2025), the balance mechanism between exploration and exploitation Yang et al. (2025); Wu et al. (2025a); Chen et al. (2025d); Wu et al. (2025a), and the cross-domain adaptation and multimodal reasoning Chen et al. (2025a); Xiao et al. (2025); Liang et al. (2025).

### 2.2 SAMPLING STRATEGIES FOR REINFORCEMENT LEARNING

Efficient sample selection is critical for the convergence and performance of LLM fine-tuning, as it directly impacts which trajectories are prioritized for learning. Several prominent sampling strategies have been proposed. Coarse-grained curriculum learning Team et al. (2025); Xie et al. (2025) gradually increases trajectory difficulty based on a competence-difficulty alignment score. LIMR Li et al. (2025b) proposes Learning Impact Measurement (LIM) to prioritize problems whose expected learning progress best matches the current model trajectory. Prioritized Sampling Team et al. (2025) weighs replay probability by TD-error or uncertainty, letting the agent reuse rare but informative transitions. Dynamic Sampling Yu et al. (2025) monitors online pass rates and resamples low-variance trajectories until their outcomes are neither 0 nor 1, reducing redundancy at the cost of extra rollouts. MCTS-structured exploration Csippán et al. (2025) leverages tree search as a policy-improvement operator to steer deep RL toward high-value regions in vast action spaces, markedly boosting sample efficiency.

### 2.3 REASONING PATTERNS OF LARGE REASONING MODELS

With the widespread adoption of RLVR, researchers have begun to investigate its effect on LLM behavior beyond simple performance metrics (Han et al., 2025; Cheng et al., 2025). Some works begin to balance direct answers with extended thought processes to alleviate the problem of overthinking (Wu et al., 2025b; Fang et al., 2025; Luo et al., 2025a; Zhang et al., 2025b). However, few explore how reasoning patterns evolve during training. To address this, Chen et al. (2025b) systematically investigates the role of RLVR for enhancing the reasoning capabilities of LLMs, discovering that their core advantage lies in optimizing the selection of existing high-success-rate reasoning patterns. Building upon this crucial insight, our work is the first to propose a training framework that explicitly leverages and optimizes this pattern selection process. To actively teach the model to pick the right pattern for each problem, thereby pushing the boundaries of LLM reasoning performance.

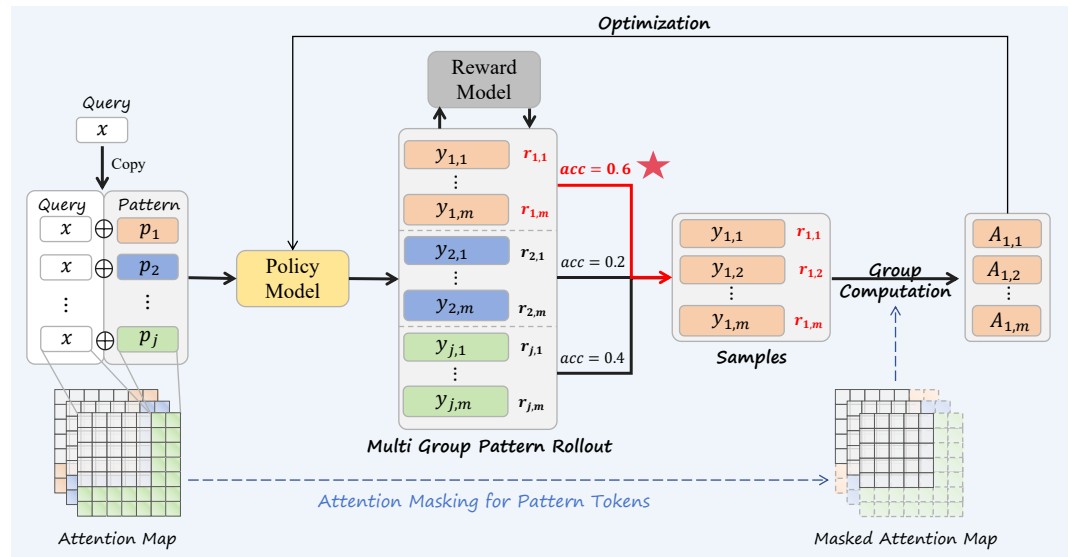

Figure 2: Overview of Group Pattern Selection Optimal (GPSO).

## 3 METHODOLOGY

In this section, we introduce Group Pattern Selection Optimization (GPSO), which teaches the model to pick the right pattern for reasoning. We first describe the preliminaries of Reinforcement Learning with Verifiable Rewards (RLVR) and then introduce our GPSO.

### 3.1 PRELIMINARIES OF RLVR

Reinforcement Learning with Verifiable Rewards (RLVR) Gao et al. (2024); Lambert et al. (2025); DeepSeek-AI et al. (2025); Team et al. (2025) refers to reinforcement learning optimization of models using rewards that can be automatically calculated using a rule-based verifier which assigns a scalar reward score to each generated response. Specifically, given a prompt $x$, the policy $\pi_\theta$ generates a reasoning trace $z$ followed by a final answer $y$. A verifier computes a reward $r = Verifier(y, y^*)$. Training proceeds via standard RL algorithms (e.g., PPO (Schulman et al., 2017) or GRPO Shao et al. (2024)) to maximize the expected verifier reward, i.e.:

$$\max_\theta \ \mathbb{E}_{z,y\sim\pi_\theta(\cdot|x)}\big[Verifier(y, y^*)\big] \tag{1}$$

where $Verifier$ is a rule-based function that compares the model output $y$ against the reference answer $y^*$ and returns a scalar score. Common instantiations include symbolic verifiers for mathematical problem solving, which check the equivalence of generated solutions against reference answers Cui et al. (2025), and sandboxed execution environments for code generation, where candidate programs are executed against unit tests to determine functional correctness (Cui et al., 2025; Gehring et al., 2025).

In this paper, we adopt Group Relative Policy Optimization (GRPO) as our reinforcement learning objective. GRPO is a PPO-like actor-only algorithm that omits the learning of a separate value function. For each prompt $x$, it samples a group of $G$ reasoning traces and answers $\{(z_i, y_i)\}_{i=1}^{G}$, each yielding a scalar reward $r_i = \text{Verifier}(y_i, y^*)$. The optimization objective is:

$$\mathcal{L}_{\text{GRPO}}(\theta) = \mathbb{E}_{x\sim\mathcal{D}}\left[\frac{1}{G}\sum_{i=1}^{G}\min\left(\frac{\pi_\theta(z_i, y_i \mid x)}{\pi_{\theta_{\text{old}}}(z_i, y_i \mid x)}A_i,\ \text{clip}\left(\frac{\pi_\theta(z_i, y_i \mid x)}{\pi_{\theta_{\text{old}}}(z_i, y_i \mid x)}, 1-\varepsilon, 1+\varepsilon\right)A_i\right)\right], \tag{2}$$

where the advantage $A_i$ is computed as follows:

$$A_i = \frac{r_i - \mu_r}{\sigma_r + \epsilon_{\text{norm}}}, \quad \mu_r = \frac{1}{G}\sum_{j=1}^{G} r_j, \quad \sigma_r = \sqrt{\frac{1}{G}\sum_{j=1}^{G}(r_j - \mu_r)^2}. \tag{3}$$

## 3.2 PATTERN-OPTIMAL REASONER

We now present our proposed method, Group Pattern Selection Optimization (GPSO), which extends RLVR with the ability to explore and learn the most effective reasoning patterns for different prompts. As shown in Figure 2, the central idea is to leverage multiple candidate patterns appended to the prompt, evaluate their effectiveness using verifier-based rewards, and then selectively update the policy with the optimal pattern while preventing overfitting to pattern-related suffix tokens through attention masking.

**Multi-Pattern Rollout.** Given a prompt $x$, we introduce a set of $n$ reasoning patterns $\{p_1, \ldots, p_n\}$. Each pattern serves as a suffix that encourages the model to follow a distinct reasoning trajectory. For each $p_j$, the policy $\pi_\theta$ samples $m$ responses:

$$\mathcal{G}_j = \{y_{j,1}, y_{j,2}, \ldots, y_{j,m}\} \sim \pi_\theta(\cdot \mid x \oplus p_j), \tag{4}$$

where $\oplus$ denotes prompt concatenation. Each response $y_{j,k}$ receives a verifier reward $r_{j,k} = Verifier(y_{j,k}, y^*)$, $y^*$ is the golden answer.

**Pattern Selection Rule.** To determine the most effective reasoning strategy, we compute the empirical accuracy of each pattern:

$$Acc(p_j) = \frac{1}{m}\sum_{k=1}^{m} \mathbf{1}[r_{j,k} = 1], \tag{5}$$

and select the optimal pattern

$$p^* = \arg\max_{p_j} Acc(p_j). \tag{6}$$

When multiple patterns achieve the same accuracy, we select the one producing the shortest valid reasoning trace $\ell(y_{j,k})$, favoring concise solutions. The responses guided by the selected pattern $p^*$ are then used to perform the subsequent policy update.

**Attention Masking for Pattern Suffix.** While suffixes $p_j$ guide exploration, we prevent the model from overfitting by masking out their contribution during gradient updates. Concretely, let $M \in \{0,1\}^{B \times (L_{\text{prompt}} + L_{\text{resp}})}$ be the attention mask, where $B$ is the batch size, $L_{\text{prompt}}$ is the maximum prompt length, and $L_{\text{resp}}$ is the maximum response length. For a given sequence, $M_{i,t} = 0$ indicates that token $t$ in instance $i$ is masked out, and $M_{i,t} = 1$ otherwise. In particular, for tokens corresponding to the appended pattern suffix, we enforce

$$M_{i,t} = 0, \quad \forall t \in \text{Idx}(p_j), \tag{7}$$

where $\text{Idx}(p_j)$ denotes the index set of token positions occupied by suffix $p_j$. This ensures that suffix tokens cannot influence the contextual representation of other tokens. Thus, patterns act as exploration scaffolds but do not directly leak into the learned policy.

**Training Objective.** Once $p^*$ is identified, we restrict optimization to its sampled group $\mathcal{G}_{p^*}$. Let $\hat{A}_{p^*,k}$ denote the group-normalized advantage, computed as in GRPO but masked such that gradient flow ignores suffix positions. Formally, the GPSO objective is:

$$\mathcal{L}_{\text{GPSO}}(\theta) = \mathbb{E}_{x \sim \mathcal{D}}\Bigg[\frac{1}{\mathcal{G}_{p^*}}\sum_{k=1}^{\mathcal{G}_{p^*}} \min\Bigg(\frac{\pi_\theta(z_k, y_k \mid x \oplus p^*)}{\pi_{\theta_{\text{old}}}(z_k, y_k \mid x \oplus p^*)} \cdot \hat{A}_{p^*,k},$$

$$\text{clip}\Bigg(\frac{\pi_\theta(z_k, y_k \mid x \oplus p^*)}{\pi_{\theta_{\text{old}}}(z_k, y_k \mid x \oplus p^*)}, 1 - \varepsilon, 1 + \varepsilon\Bigg) \cdot \hat{A}_{p^*,k}\Bigg)\Bigg]. \tag{8}$$

Here, $\hat{A}_{p^*,k}$ is computed by normalizing rewards within $\mathcal{G}_{p^*}$, and gradients are masked to exclude suffix tokens. In this way, GPSO leverages pattern-based exploration to discover effective reasoning trajectories while maintaining a clean separation between exploration scaffolds and the policy itself.

## 4 EXPERIMENTAL METHODOLOGY

In this section, we describe the datasets, evaluation metrics, baselines, and implementation details.

**Dataset.** For the training set, we use DAPO-Math-17K dataset Yu et al. (2025), which is a curated collection of approximately 17,000 competition-level math problems. For testing, we evaluate the effectiveness of GPSO on AIME2024 Beeching et al. (2024), AIME2025 Ye et al. (2025), MATH-500 Hendrycks et al. (2021), and GPQA datasets (Rein et al., 2023).

**Evaluation Metrics.** We follow previous work Chen et al. (2021); Li et al. (2024); Wang et al. (2024); Yang et al. (2024); Luo et al. (2023) and we use `Pass@k` Chen et al. (2021) to evaluate the effectiveness of different models. In this work, we set $k = 1$. The `Pass@1` accuracy is averaged over 4 samples per problem on all benchmarks.

**Baselines.** We compare GPSO with several LRMs, such as DeepSeek-R1-Distill-Qwen-1.5B/14B DeepSeek-AI et al. (2025), DeepScaleR-1.5B-Preview Luo et al. (2025b), Light-R1-7B-DS Wen et al. (2025a), AReal-boba-RL-7B Fu et al. (2025). DeepScaleR-1.5B-Preview is further trained starting from DeepSeek-R1-Distill-Qwen-1.5B, while Light-R1-7B-DS and AReal-boba-RL-7B are further trained from DeepSeek-R1-Distill-Qwen-7B.

**Implementation Details.** In our experiments, we apply GPSO to four LRMs: Nemotron-Research-Reasoning-Qwen-1.5B Liu et al. (2025), DeepSeek-R1-Distill-Qwen-7B DeepSeek-AI et al. (2025), DeepSeek-R1-Distill-Llama-8B DeepSeek-AI et al. (2025), and Qwen3-8B (Thinking) (Team, 2025). During training, we use Verl framework Sheng et al. (2024) and apply GRPO as the RL algorithm to implement GPSO. For hyperparameters, we set the batch size and mini-batch size to $64$, and for each problem, we rollout $8$ responses using four patterns: Direct Solution, Reflection and Verification, Exploration of Multiple Solutions, and Adaptive. The maximum lengths for prompts and responses are $1,024$ and $16,384$ tokens, respectively. The learning rate is set to $1e-6$, and we adopt the AdamW optimizer for the policy model. All experiments were conducted on $8 \times$ NVIDIA H20 96GB GPUs. During testing, we set the temperature to $0.6$. The maximum generation length is set to $32,768$ tokens for AIME 2024/2025 and $16,384$ tokens for MATH-500 and GPQA. All evaluations are conducted under the zero-shot setting.

## 5 EVALUATION RESULTS

In this section, we present the evaluation results for GPSO. Our evaluation includes a comprehensive analysis of the overall performance, ablation studies to assess the contribution of key components, and insights into how GPSO enhances reasoning performance across a variety of tasks.

### 5.1 OVERALL PERFORMANCE

The overall performance of GPSO is shown in Table 1. Across different model backbones, applying GPSO consistently improves performance. Nemotron-Research-Reasoning-Qwen-1.5B improves its average score from 55.4 to 58.0 (+2.6%), while DeepSeek-R1-Distill-Qwen-7B increases from 55.6 to 58.7 (+3.1%). Similarly, DeepSeek-R1-Distill-Llama-8B improves from 51.4 to 54.6 (+3.2%). Notably, Qwen3-8B (Thinking) further benefits from GPSO, achieving the best overall average of 75.3. These results indicate that GPSO is model-agnostic and provides stable gains. On individual benchmarks, the improvements brought by GPSO mainly come from challenging reasoning tasks such as AIME 2024 and AIME 2025. Across all four models, GPSO yields an average gain of 4.0 points on AIME2024 and 2.7 points on AIME2025. Moreover, although GPSO is trained solely on mathematical data, it demonstrates strong generalization across domains, achieving an average improvement of 2.1 points on GPQA. These results confirm that GPSO offers a plug-and-play enhancement to existing RLVR training pipelines, with consistent gains across both weak and strong LLMs.

Table 1: Overall performance of Group Pattern Selection Optimization (GPSO).

| Model | AIME2024 | AIME2025 | MATH500 | GPQA | Avg. |
|---|---|---|---|---|---|
| **DeepSeek-R1-Distill-Qwen-1.5B** | 30.0 | 20.0 | 84.7 | 33.8 | 42.1 |
| **DeepScaleR-1.5B-Preview** | 40.2 | 28.5 | 87.8 | 32.3 | 47.2 |
| **Light-R1-7B-DS** | 57.7 | 46.4 | 91.1 | 47.2 | 60.6 |
| **AReal-boba-RL-7B** | 62.7 | 49.4 | 93.8 | 48.0 | 63.5 |
| **DeepSeek-R1-Distill-Qwen-14B** | 70.4 | 50.0 | 92.4 | 59.5 | 68.1 |
| **Nemotron-Research-Reasoning-Qwen-1.5B** | 53.3 | 35.8 | 92.1 | 40.5 | 55.4 |
| **+ GPSO** | **58.3** | **37.5** | **93.1** | **43.2** | **58.0** |
| **DeepSeek-R1-Distill-Qwen-7B** | 48.3 | 33.3 | 93.2 | 47.6 | 55.6 |
| **+ GPSO** | **53.3** | **40.0** | **93.5** | **47.9** | **58.7** |
| **DeepSeek-R1-Distill-Llama-8B** | 44.2 | 27.5 | 88.1 | 46.0 | 51.4 |
| **+ GPSO** | **49.2** | **29.2** | **90.2** | **50.0** | **54.6** |
| **Qwen3-8B (Thinking)** | 76.7 | 67.5 | 96.0 | 58.0 | 74.5 |
| **+ GPSO** | **77.5** | **68.3** | **96.1** | **59.2** | **75.3** |

Table 2: Ablation Studies. We evaluate the impact of removing each component in GPSO: Multi-Pattern Rollout (MPR), Optimal Pattern Selection (OPS), Masking Pattern Tokens (Mask), and the KL penalty (KL). ✓ indicates the component is enabled, while ✗ indicates it is disabled.

| Model | MPR | OPS | Mask | KL | AIME2024 | AIME2025 | GPQA | Avg |
|---|---|---|---|---|---|---|---|---|
| Nemotron-Qwen-1.5B | - | - | - | - | 53.3 | 35.8 | 40.5 | 43.2 |
| Nemotron-Qwen-1.5B-GPSO | ✓ | ✓ | ✓ | ✓ | **58.3** | **37.5** | **43.2** | **46.3** |
| w/o KL | ✓ | ✓ | ✓ | ✗ | 54.2 | 36.7 | 40.8 | 43.9 |
| w/o Multi-Pattern Rollout | ✗ | ✗ | ✗ | ✓ | 49.2 | 33.3 | 40.5 | 41.0 |
| w/o Optimal Pattern Selection | ✓ | ✗ | ✓ | ✓ | 53.3 | 35.8 | 40.0 | 43.1 |
| w/o Mask Pattern Tokens | ✓ | ✓ | ✗ | ✓ | 56.7 | 33.3 | 40.3 | 43.4 |
| DeepSeek-R1-Distill-Qwen-7B | - | - | - | - | 48.3 | 33.3 | 47.6 | 43.2 |
| DeepSeek-R1-Distill-Qwen-7B-GPSO | ✓ | ✓ | ✓ | ✓ | **53.3** | **40.0** | 47.9 | **47.1** |
| w/o KL | ✓ | ✓ | ✓ | ✗ | 52.5 | 36.7 | 46.3 | 45.2 |
| w/o Multi-Pattern Rollout | ✗ | ✗ | ✗ | ✓ | 50.8 | 35.8 | 47.7 | 44.8 |
| w/o Optimal Pattern Selection | ✓ | ✗ | ✓ | ✓ | 50.8 | 38.3 | **50.3** | 46.5 |
| w/o Mask Pattern Tokens | ✓ | ✓ | ✗ | ✓ | 51.7 | 38.3 | 45.7 | 45.2 |

## 5.2 ABLATION STUDIES

To further investigate the individual contributions of the key components in GPSO, we conduct a series of ablation experiments. As shown in Table 2 and Figure 3, we evaluate the model under several settings: (1) removing the KL penalty, (2) excluding the Multi-Pattern Rollout mechanism, (3) disabling the Optimal Pattern Selection, and (4) not masking the Pattern Tokens. From the results and training accuracy curves on AIME2024 and AIME2025, we observe that removing any of these components leads to a noticeable performance degradation.

We summarize our key observations as follows. First, the **removal of the KL penalty** consistently hurts performance across both models and datasets. As shown in Figure 3, the training curves without KL (yellow triangles) remain persistently below the GPSO baseline (purple triangles), indicating that the KL regularization effectively stabilizes training and prevents overfitting to spurious patterns. This is also reflected in Table 2, where removing KL leads to a drop of 2.4 points (Nemotron) and 1.9 points (DeepSeek-R1) on average.

Second, excluding the **Multi-Pattern Rollout mechanism**(blue squares) leads to the most significant performance degradation. Without rollout, the model struggles to explore diverse reasoning paths and quickly plateaus. The average performance drops by 5.3 points (Nemotron) and 2.3 points (DeepSeek-R1), highlighting the critical role of this component in guiding exploration.

Third, turning off **Optimal Pattern Selection** results in a moderate but consistent decrease. Although the Multi-Pattern Rollout still runs, the lack of selection prevents the model from reinforcing

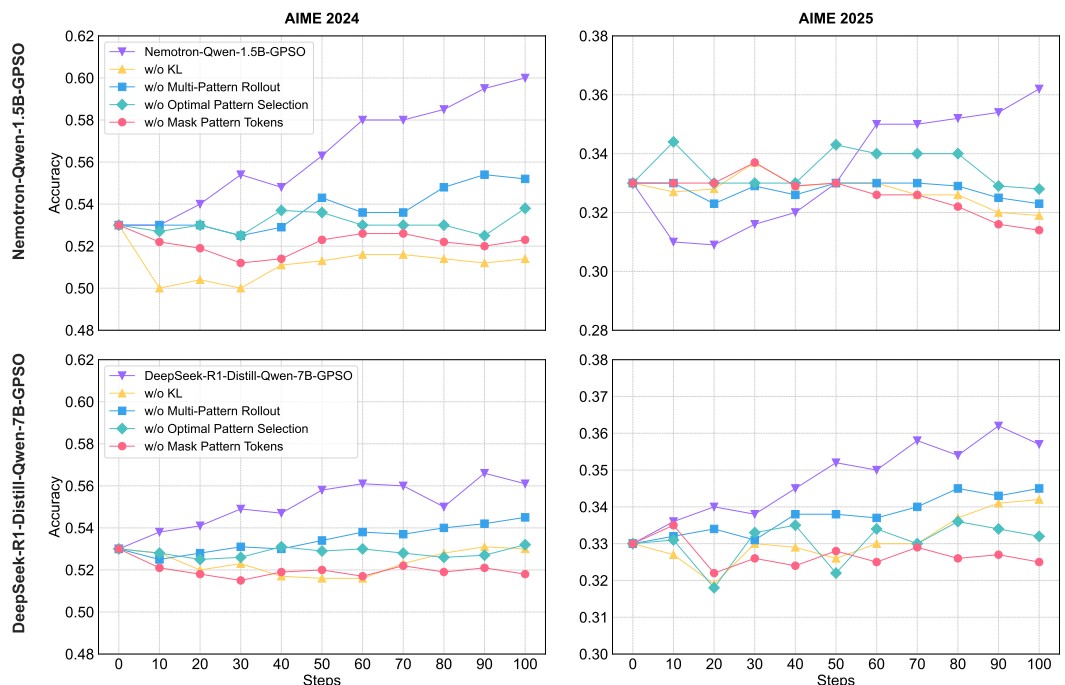

Figure 3: Training Accuracy Curves On AIME2024 and AIME2025.

Table 3: Effectiveness of Reasoning Pattern Selection with GPSO

| Model | Pattern | AIME2024 | AIME2025 | MATH500 | GPQA |
|---|---|---|---|---|---|
| **Nemotron-Qwen-1.5B** | - | 53.3 | 33.3 | 92.1 | 40.5 |
| | Direct Solution | 41.7 | 27.5 | 90.1 | 39.3 |
| | Reflection and Verification | 47.5 | 30.8 | **92.2** | **41.3** |
| | Explore Multiple Solutions | **55.0** | **34.2** | 91.7 | 38.6 |
| **Nemotron-Qwen-1.5B-GPSO** | - | **58.3** | **37.5** | **93.1** | **43.2** |
| | Direct Solution | 49.2 | 33.3 | 91.1 | 40.5 |
| | Reflection and Verification | 53.3 | 35.8 | 91.6 | 40.0 |
| | Explore Multiple Solutions | 56.7 | 33.3 | 91.3 | 40.3 |
| **DeepSeek-R1-Distill-Qwen-7B** | - | 48.3 | 33.3 | **93.2** | **47.6** |
| | Direct Solution | 43.3 | 26.7 | 88.1 | 44.8 |
| | Reflection and Verification | **50.0** | 33.3 | 92.6 | **47.6** |
| | Explore Multiple Solutions | 48.3 | **36.7** | 90.3 | 47.5 |
| **DeepSeek-R1-Distill-Qwen-7B-GPSO** | - | 53.3 | 40.0 | 93.5 | 47.9 |
| | Direct Solution | 45.0 | 36.7 | 90.2 | 46.5 |
| | Reflection and Verification | 49.2 | 38.3 | 92.8 | 47.4 |
| | Explore Multiple Solutions | 51.7 | 39.2 | 92.2 | 47.0 |

high-quality patterns, leading to noisier supervision. This is most noticeable on AIME2025, where accuracy deteriorates by 1.7 points on both models.

Lastly, we observe that **Masking Pattern Tokens** also plays a subtle but meaningful role. Without this masking, the model has access to hard-coded pattern identifiers, which may introduce undesirable shortcuts during learning. Both Table 2 and Figure 3 show that disabling masking results in slower convergence and slightly worse final performance, suggesting that overfitting to pattern identity is more detrimental in harder, unfamiliar tasks.

### 5.3 GPSO LEARNS TO PICK THE RIGHT PATTERN FOR REASONING

As shown in Table 3, GPSO enables both models to dynamically apply the most suitable reasoning pattern per instance, outperforming all fixed-pattern baselines. Without GPSO, no single reasoning mode consistently dominates across benchmarks. For example, Nemotron-Qwen-1.5B performs

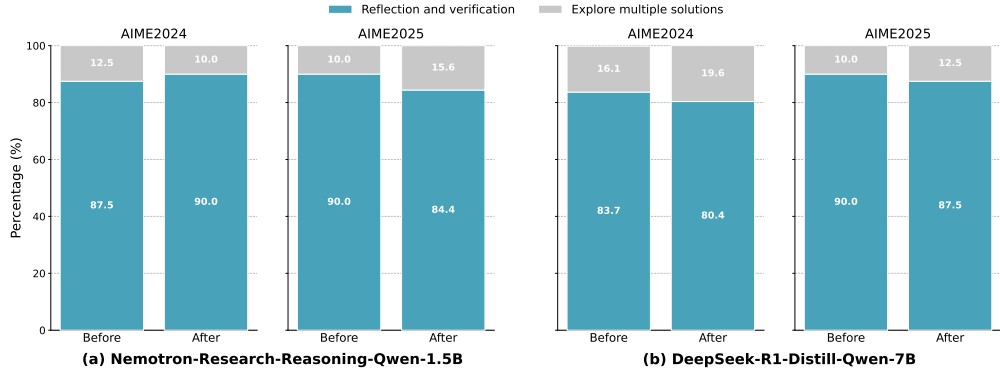

Figure 4: Pattern Usage Distribution Before and After GPSO Training

best with Explore Multiple Solutions on AIME2024 and AIME2025, but achieves higher scores with Reflection and Verification on MATH500 and GPQA. DeepSeek-R1-Distill-Qwen-7B shows similar variability.

In contrast, GPSO-trained models achieve the highest scores across all benchmarks using the default decoding strategy—surpassing even the best fixed-pattern results. This demonstrates that GPSO can effectively learn to adaptively combine reasoning strategies based on the problem type, leading to more generalizable and robust performance.

## 5.4 DISTRIBUTION OF REASONING PATTERNS BEFORE AND AFTER GPSO TRAINING.

As shown in Figure 4, we analyze the distribution of reasoning patterns selected on the AIME2024 and AIME2025 datasets, both before and after applying GPSO. The results confirm that GPSO enables models to learn an adaptive, problem-dependent policy rather than converging to a single fixed strategy.

For Nemotron-Research-Reasoning-Qwen-1.5B (Figure 4(a)), we observe a clear task-specific adjustment. On AIME2024, the model further strengthens its preference for the Reflection and Verification pattern, increasing its usage from 87.5% to 90.0%. In contrast, on the more challenging AIME2025, the model shifts towards Explore Multiple Solutions, increasing its usage from 10.0% to 15.6%. This indicates that GPSO guides the model to adopt more exploratory strategies when the problem requires it.

A similar trend is observed with DeepSeek-R1-Distill-Qwen-7B (Figure 4(b)). On AIME2024, the share of Explore Multiple Solutions rises from 16.1% to 19.6%, and on AIME2025, from 10.0% to 12.5%. These shifts further highlight GPSO's ability to learn a meta-policy that adjusts the invocation probabilities of different reasoning strategies based on task characteristics.

## 6 CONCLUSION

In this work, we propose GPSO, a novel training paradigm that enables language models to select optimal reasoning patterns per instance dynamically. By combining multi-pattern exploration, verifier-guided supervision, and gradient-masked updates, GPSO teaches the model to internalize reasoning strategies without relying on explicit prompts. Experiments on multiple benchmarks demonstrate that GPSO consistently improves performance across models and tasks, particularly on challenging datasets that require reasoning. Our results highlight the effectiveness of adaptive pattern selection in enhancing both accuracy and generalization of LLM reasoning.

ETHICS STATEMENT

We affirm that this work adheres to the ICLR Code of Ethics . This study does not involve human subjects, sensitive demographic information, or data collected from vulnerable populations. All datasets used are publicly available and commonly used in the field. The methods proposed in this paper do not pose foreseeable risks of misuse or harm. We are not aware of any discrimination, bias, or fairness concerns introduced by our methodology. No conflicts of interest or undisclosed sponsorships are involved. Any software released as part of this research will comply with relevant privacy, security, and licensing requirements.

REPRODUCIBILITY STATEMENT

We have made extensive efforts to ensure the reproducibility of our results. A detailed description of our methodology architecture and training pipeline is provided in Section 3. The evaluation metrics, experimental setup, and complete hyperparameter configurations are described in Section 4. All datasets used in our study are either publicly available or explicitly referenced in the text. To further facilitate reproducibility, we include the full prompt templates and representative input-output examples in Appendix B.

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

# A Distribution of Reasoning Patterns Across Domains

To support our analysis, we sample 1,000 reasoning trajectories from each model for the mathematics or science domains, respectively. Each response is annotated into one of five high-level reasoning categories: Direct Solution, Explore Multiple Solutions, Reflection and Verification, Analogy, and Reverse Thinking.

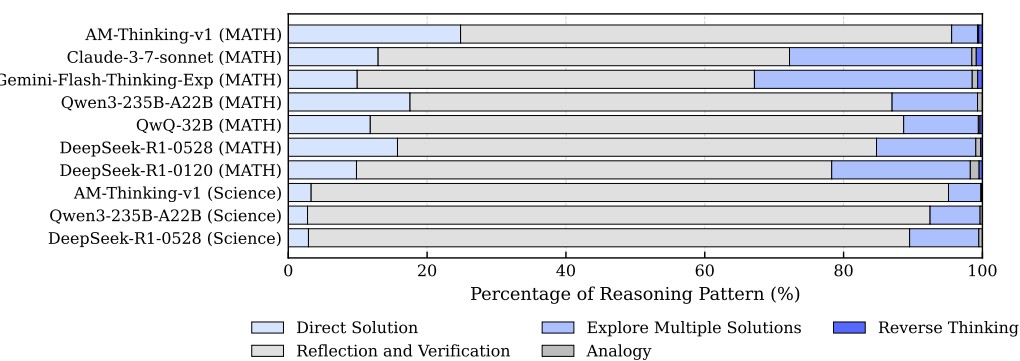

Figure 5: Distribution of reasoning patterns used by various LLMs on MATH and Science tasks

Figure 5 provides a detailed breakdown of the percentage distribution of reasoning patterns exhibited by each model. We observe clear trends—such as the dominance of Reflection and Verification in most models, particularly in the science domain, and the relatively lower adoption of Explore Multiple Solutions or Reverse Thinking, even for state-of-the-art models like Claude and Gemini. These results underscore the tendency of LLMs to default to a small subset of reasoning strategies, despite their architectural capacity for diverse reasoning.

# B Prompts

## B.1 Full Prompts for Pattern Analysis

> **Full Prompt for Responses Classification**
>
> I will provide you with a problem and its solution. Please analyze the strategy used in this solution.
> When solving problems, people may use the following strategies:
> - **Direct solution**: Solve the problem step by step, without reflection or verification, and finally give the answer.
> - **Reflection and verification**: Reflect on the solution, verify the solution, if errors are found, rethink the problem, solve the problem again, and finally give the answer.
> - **Explore multiple solutions**: Explore multiple solutions, select the optimal solution or synthesize multiple solutions to give the final answer.
> - **Others**: The strategy used in the solution is not among the above strategies.
> Please analyze which strategy this solution uses, select the single strategy you believe is most appropriate.
> Problem: {question}
> Solution: {solution}
> In your response, you first conduct the analysis and at the end use the `<strategy> your strategy </strategy>` tag to describe the strategy used in the solution. Please note that the tags should only contain the strategy and no other content.
> **Strategy**:

## B.2   PROMPTS FOR PATTERN REASONING

---

**Prompt Example for Pattern no**

Convert the point $(0, 3)$ in rectangular coordinates to polar coordinates. Enter your answer in the form $(r, \theta)$, where $r > 0$ and $0 \le \theta < 2\pi$.
Present the answer in LaTeX format: `\boxed{Your answer}`

---

**Prompt Example for Pattern short_cot**

Convert the point $(0, 3)$ in rectangular coordinates to polar coordinates. Enter your answer in the form $(r, \theta)$, where $r > 0$ and $0 \le \theta < 2\pi$.
Present the answer in LaTeX format: `\boxed{Your answer}`
In your response, provide the final answer in a concise manner. Do not need to self-reflect or self-criticize your reasoning process.

---

**Prompt Example for Pattern explore_multiple_solutions**

Convert the point $(0, 3)$ in rectangular coordinates to polar coordinates. Enter your answer in the form $(r, \theta)$, where $r > 0$ and $0 \le \theta < 2\pi$.
Present the answer in LaTeX format: `\boxed{Your answer}`
In your response, explore multiple solutions to the problem, and then provide the final answer based on the these solutions.

---

**Prompt Example for Pattern reflection_and_verification**

Convert the point $(0, 3)$ in rectangular coordinates to polar coordinates. Enter your answer in the form $(r, \theta)$, where $r > 0$ and $0 \le \theta < 2\pi$.
Present the answer in LaTeX format: `\boxed{Your answer}`
In your response, reflect on your reasoning process and verify the correctness of your solution before providing the final answer. If you find any errors in your reasoning, correct them and provide the final answer.

---

## B.3   SPECIAL PROMPT FOR GPQA EVALUATION

---

**Special Prompt for GPQA Evaluation**

You **MUST** follow these output rules strictly:

1. Think as you wish, but the **FINAL** line must be exactly one LaTeX box with a single letter among A, B, C, D.

2. The final line format **MUST** be: `\boxed{A}` or `\boxed{B}` or `\boxed{C}` or `\boxed{D}` (uppercase only).

3. Do **NOT** add extra text, spaces, punctuation, or anything else after the box.

4. If you mention option text, reference the text exactly as shown in the problem.

---

### B.4 PROMPT EXAMPLES FOR EVALUATION

---

**Full Prompt and response for Evaluation**

```
<|im_start|>system
```
A conversation between User and Assistant. The User asks a question, and the Assistant solves it. The Assistant first thinks through the reasoning process internally, then provides the user with the answer.
Example format:
`<think>`...(Reasoning process here, attempt until you get the final answer)`</think>`
`<answer>`...(Summarize the final solution, and present the answer).`</answer>`
User: {`prompt`}.
Assistant: `<think>`
```
<|im_end|>
```

```
<|im_start|>user
```
Convert the point $(0, 3)$ in rectangular coordinates to polar coordinates. Enter your answer in the form $(r, \theta)$, where $r > 0$ and $0 \leq \theta < 2\pi$.
Present the answer in LaTeX format: `\boxed{Your answer}`
```
<|im_end|>
```

```
<|im_start|>assistant
<think>...</think>
```
To convert the rectangular coordinates $(0, 3)$ to polar coordinates $(r, \theta)$, we use the following standard conversion formulas:

$$r = \sqrt{x^2 + y^2}, \quad \theta = \tan^{-1}\left(\frac{y}{x}\right)$$

— **Step 1: Compute $r$**
Given the point $(x, y) = (0, 3)$, we substitute into the formula for $r$:

$$r = \sqrt{0^2 + 3^2} = \sqrt{9} = 3$$

So, $r = 3$.
— **Step 2: Compute $\theta$**
Next, we find $\theta$, the angle made with the positive $x$-axis. The formula is:

$$\theta = \tan^{-1}\left(\frac{y}{x}\right)$$

However, here $x = 0$, so $\frac{y}{x}$ is undefined. This means the point lies on the **positive $y$-axis**.
In standard polar coordinate representation, the positive $y$-axis corresponds to an angle of:

$$\theta = \frac{\pi}{2}$$

This is the angle that places the point directly above the origin on the positive $y$-axis.
— **Step 3: Verify the Result**
To confirm, we can convert back from polar coordinates to rectangular coordinates using:

$$x = r\cos\theta, \quad y = r\sin\theta$$

Substituting $r = 3$, $\theta = \frac{\pi}{2}$:

$$x = 3\cos\left(\frac{\pi}{2}\right) = 3 \cdot 0 = 0, \quad y = 3\sin\left(\frac{\pi}{2}\right) = 3 \cdot 1 = 3$$

This matches the original rectangular coordinates $(0, 3)$, verifying our conversion is correct.
— **Final Answer**

$$\boxed{(3, \frac{\pi}{2})}$$

```
<|im_end|>
```

## C  THE USE OF LARGE LANGUAGE MODELS (LLMs)

Large Language Models (LLMs), specifically ChatGPT, are used solely for language polishing, improving the clarity and readability of the manuscript, and assisting in the retrieval of related work during the literature review stage. All scientific content, including the research idea, methodology, experiments, analysis, and conclusions, is conceived and written by the authors without assistance from LLMs.

The authors take full responsibility for the content of this paper, including all text generated with the aid of LLMs.

