# OpenReview forum: "Group Pattern Selection Optimal: Let LRMs Pick the Right Pattern for Reasoning"
_ICLR.cc/2026/Conference — ICLR 2026 Conference Withdrawn Submission_

### Official Review · Reviewer_p289 · 2025-10-27

**Soundness:** 2
**Presentation:** 3
**Contribution:** 3
**Rating:** 6
**Confidence:** 4

**Summary:**

This paper identifies that Large Reasoning Models (LRMs) often default to dominant, sub-optimal reasoning patterns. It introduces Group Pattern Selection Optimization (GPSO), a reinforcement learning framework extending GRPO. The method operates by performing multi-pattern rollouts, where a single problem is prompted with several different "reasoning pattern suffixes". A verifier signal is used to identify the most effective pattern for that specific problem, and the model's policy is then updated using only the trajectories from that optimal pattern group. GPSO employs attention masking to prevent the model from overfitting to the explicit pattern suffixes, thereby forcing it to learn an intrinsic mapping from the problem to the optimal reasoning strategy. Experiments on math and science benchmarks demonstrate consistent, though often modest, performance gains.

**Strengths:**

- The motivation is clear. Figure 1 demonstrates that different reasoning patterns yield different accuracies for the same problem and that current RLVR pipelines can bias models toward a single, sub-optimal dominant pattern.

- The proposed training mechanism, GPSO, is general and well-designed. It builds on GRPO by adding multi-pattern rollouts, best-pattern selection, and attention masking, which could potentially be integrated into various RLVR-style pipelines.

- The contribution of the reasoning pattern optimization, using masking for pattern prompting invariance and a best-group update to force the model to learn an internal reasoning pattern policy, is a valuable idea for improving the quality of intermediate reasoning tokens generated by the model.

- The method shows consistent performance gains, particularly on weaker to midsize models (e.g., ~2–3 point average gains on 1.5B–8B parameter models) on difficult math benchmarks.

**Weaknesses:**

1. The pipeline’s absolute effectiveness remains unclear because the trained model still trails very much behind the oracle “Best” per-question pattern upper bound from the analysis in Figure 1; the gap (e.g. 90% best on AIME2024 with Qwen3 thinking compared to the achieved 77.5% which is only less than 1% gain compared to the baseline 76.7% and trailing behind even some reasoning pattern prompting methods) suggests the bottleneck is only slightly addressed.

2. The training objective relies on a "hard" best-pattern selection. The method always picks the single best pattern group (by highest verified accuracy) and completely ignores all other patterns. This discards potentially useful training signals from other rollouts. An ablation exploring "soft" inclusion (e.g., weighting each pattern group by its verifier score) would be useful to support the hard best-only design.

3. There is a lack of statistical significance reporting. Evaluation relies on Pass@1 averaged over 4 samples per problem, decoded at a non-zero temperature (T=0.6). Results are reported as single numbers without variance or confidence intervals. This reduces the impact of the findings, especially since several gains on stronger models are less than 1 point (e.g., +0.8 on Qwen3-8B) and could be statistical noise.

4. Despite discussing patterns like “employing tools,” the experiments are confined to math/science QA with textual reasoning. There is no evaluation on tasks that require external tools (e.g., code execution, retrieval, calculators), task decomposition, or search, so it is unclear whether pattern effects and GPSO’s gains persist in those domains.

**Questions:**

1. Can you report variance or confidence intervals for the results in Tables 1?

2. How does GPSO's performance compare to a simpler inference-time selection baseline? For example, prompting the base model with all n (3) patterns, sampling m times, and  using majority voting for example to select the final answer? This would help isolate the gains from the RL training itself versus the multi-pattern ensemble.

3. What is your intuition for why a "soft inclusion" of other reasoning patterns (e.g., a weighted mix based on verifier scores) would not outperform the "best-only" hard selection strategy?

4. Have you measured the intrinsic problem -> pattern mapping accuracy (e.g., probability of generating the correct pattern without an explicit prompt) before and after GPSO training?

---

### Official Review · Reviewer_49U5 · 2025-11-02

**Soundness:** 1
**Presentation:** 1
**Contribution:** 1
**Rating:** 2
**Confidence:** 5

**Summary:**

The authors present a work to incorporate a set of handcrafted reasoning patterns when fine-tuning LRMs with RLVR. To do so, they prompt the model with the pattern in the prompt, keeping the best pattern according to success rate for the advantage calculation. They additionally mask out the pattern suffix to mask out their contribution during updates.

**Strengths:**

- Evaluate the performance of the approach in a variety of math reasoning domains
- Perform an ablation study of each component of their method to show the contribution

**Weaknesses:**

- GRPO with a sample equivalent N (e.g N = num_patterns x num_samples per pattern) not compared against as a baseline
- Patterns are handcrafted per task and can be viewed as a basic form of prompt optimization[1,2,3,4, 5], which has a rich history of approaches and is not compared against. Thus unclear about novelty of proposed approach.

References
[1] DSPy: Compiling Declarative Language Model Calls into Self-Improving Pipelines
[2] RLPrompt: Optimizing Discrete Text Prompts with Reinforcement Learning
[3] GEPA: Reflective Prompt Evolution Can Outperform Reinforcement Learning
[4] A Systematic Survey of Automatic Prompt Optimization Techniques
[5] Prefix-Tuning: Optimizing Continuous Prompts for Generation

**Questions:**

See Weaknesses above.

---

> ### Author Response · Authors · 2025-11-16
> **Response to Reviewer 49U5**
>
> We sincerely thank you for the constructive feedback. We address each concern below and will incorporate these clarifications in our revision. We appreciate it very much if you kindly raise the scores when the concerns are addressed.
>
> ## Reply to W1: Baseline with Larger Number of Rollouts
>
> Thank you for this important concern. The baseline you requested is actually already included in **Table 2 as "w/o Multi-Pattern Rollout"**. We appreciate this feedback and will make this comparison more explicit in the revision.
>
> In "w/o Multi-Pattern Rollout" ablation setting, we disable pattern-specific rollouts and instead sample the same total number of responses per problem (rollouts = num_patterns × num_samples_per_pattern) without any pattern conditioning. This provides a direct comparison under identical computational budgets.
>
> Experimental Setup:
>
>  - **GPSO**: 4 patterns × 8 samples = **32 rollouts per problem** (with pattern conditioning), however, only the 8 rollouts of the optimal pattern will be selected for update.
>  - **w/o Multi-Pattern Rollout**: **32 rollouts per problem** (standard GRPO, no pattern conditioning)
>
> The detailed comparison results are shown in the table below.
>
> | Model                       | AIME2024 | AIME2025 | GPQA     | Avg.     |
> | --------------------------- | -------- | -------- | -------- | -------- |
> | Nemotron-Qwen-1.5B          |          | 35.8     | 40.5     | 43.2     |
> | + GRPO                      | 49.2     | 33.3     | 40.5     | 41.0     |
> | + GPSO                      | **58.3** | **37.5** | **43.2** | **46.3** |
> | DeepSeek-R1-Distill-Qwen-7B | 48.3     | 33.3     | 47.6     | 43.2     |
> | + GRPO                      | 50.8     | 35.8     | 47.7     | 44.8     |
> | + GPSO                      | **53.3** | **40.0** | **47.9** | **47.1** |
>
> ## Reply to W2: Response regarding novelty and relation to prompt optimization
>
> Thank you for question.
>
> While GPSO incorporates handcrafted patterns as exploration scaffolds, its core contribution is *not* prompt optimization in the conventional sense. Prior work on prompt optimization [1–5]—focuses on **searching or optimizing prompts** to elicit better model behaviors. In contrast, our patterns are *not* optimized, tuned, or selected as prompts. Instead, they serve as **latent reasoning modes** that already exist within LRMs (e.g., Direct → Verification → Exploration), and GPSO aims to *learn the intrinsic mapping from problem → optimal pattern* under RLVR.
>
> More importantly, GPSO makes two conceptual departures from prior prompt optimization work:
>
> 1. **Patterns are not part of the learned policy.**
>    Through attention masking, suffix tokens are removed from gradient flow, preventing the model from learning to mimic or depend on explicit pattern prompts. This distinguishes GPSO from any form of prompt optimization, prefix tuning, or prompt search, where the optimized prompt *must* influence—or become embedded in—the model policy.
> 2. **GPSO optimizes pattern–problem alignment, not pattern content.**
>    Our method trains the model to internalize *when* to use different reasoning strategies, rather than *how to write a better prompt*. The selectable patterns remain fixed; the learned component is the model’s **meta-reasoning policy** that decides which reasoning mode to activate without explicit cues at inference time. This constitutes a different problem formulation from classical prompt optimization, which searches for prompt text to enhance performance under a static reasoning mode.
>
> For these reasons, GPSO is fundamentally not a prompt optimization method, but a reinforcement-learning framework for teaching LRMs **adaptive reasoning-mode selection**.

---

> > ### Author Response · Authors · 2025-11-21
> > **Response to Reviewer 49U5**
> >
> > Dear Reviewer,
> >
> > I hope this message finds you well. We have addressed all the concerns you raised. We believe the revisions have significantly strengthened the paper.
> >
> > We would be grateful for any update on the review status at your convenience. Thank you very much for your valuable feedback and time.

---

### Official Review · Reviewer_Hgif · 2025-11-02

**Soundness:** 2
**Presentation:** 2
**Contribution:** 2
**Rating:** 2
**Confidence:** 5

**Summary:**

GPSO is a reinforcement-learning framework that lets a model try several high-level reasoning patterns on the same problem and then pick the best one using verifier signals. It extends GRPO with multi-pattern rollouts and an attention-masking trick so the model doesn’t just memorize pattern suffixes but actually learns when to use which pattern. Their analysis shows current models often choose a suboptimal pattern, and GPSO learns a problem→pattern mapping that reduces this mismatch. The authors also conduct some experiments to show the performance of their method.

**Strengths:**

1.	The method explicitly models diverse reasoning patterns instead of forcing one dominant pattern.
2.	The mehtod wses verifier signals to select the best pattern per problem, improving sample efficiency and final accuracy.

**Weaknesses:**

1.	The paper should include a baseline that uses the original GRPO but with a comparable (i.e., larger) number of rollouts/samples per batch, to show that GPSO’s gains are not just from extra sampling.
2.	The procedure for constructing the pattern set needs to be spelled out: how are the patterns obtained, filtered, and validated, and what evidence do we have that this set is sufficiently diverse for the target domains?
3.	The motivation for “selecting” a single best pattern is unclear; please clarify why optimizing all correct patterns isn’t preferable, since learning to solve a problem via multiple patterns may improve robustness and generalization.
4.	Please discuss applicability to stronger/larger models: do they still benefit from explicit pattern conditioning, or do their naturally diverse trajectories make GPSO less useful at scale?

**Questions:**

Please refer weakness

---

> ### Author Response · Authors · 2025-11-16
> **Response to Reviewer Hgif**
>
> We sincerely thank you for the constructive feedback. We address each concern below and will incorporate these clarifications in our revision. We appreciate it very much if you kindly raise the scores when the concerns are addressed.
>
> ## Reply to W1: Baseline with Larger Number of Rollouts
>
> Thank you for this important concern. The baseline you requested is actually already included in **Table 2 as "w/o Multi-Pattern Rollout"**. We appreciate this feedback and will make this comparison more explicit in the revision.
>
> In "w/o Multi-Pattern Rollout" ablation setting, we disable pattern-specific rollouts and instead sample the same total number of responses per problem (rollouts = num_patterns × num_samples_per_pattern) without any pattern conditioning. This provides a direct comparison under identical computational budgets.
>
> Experimental Setup:
>
>  - **GPSO**: 4 patterns × 8 samples = **32 rollouts per problem** (with pattern conditioning), however, only the 8 rollouts of the optimal pattern will be selected for update.
>  - **w/o Multi-Pattern Rollout**: **32 rollouts per problem** (standard GRPO, no pattern conditioning)
>
> The detailed comparison results are shown in the table below.
>
> | Model                       | AIME2024 | AIME2025 | GPQA     | Avg.     |
> | --------------------------- | -------- | -------- | -------- | -------- |
> | Nemotron-Qwen-1.5B          | 53.3     | 35.8     | 40.5     | 43.2     |
> | + GRPO                      | 49.2     | 33.3     | 40.5     | 41.0     |
> | + GPSO                      | **58.3** | **37.5** | **43.2** | **46.3** |
> | DeepSeek-R1-Distill-Qwen-7B | 48.3     | 33.3     | 47.6     | 43.2     |
> | + GRPO                      | 50.8     | 35.8     | 47.7     | 44.8     |
> | + GPSO                      | **53.3** | **40.0** | **47.9** | **47.1** |
>
> ## Reply to W2: Pattern Set Construction Procedure
>
> Thank you for highlighting the need for greater clarity on our pattern construction methodology. We provide a detailed explanation below and have included comprehensive documentation in **Appendix A**.
>
> We analyze 10,000 reasoning trajectories from 7 state-of-the-art LRMs across math and science domains using GPT-4o to systematically identify high-level reasoning strategies. From this analysis, we discover that three patterns—**Direct Solution** , **Reflection and Verification** , and **Explore Multiple Solutions**—account for **over 98%** of all observed reasoning behaviors. The remaining infrequent patterns (Analogy and Reverse Thinking) are consolidated into a fourth **"Adaptive"** category, which allows the model to freely choose its approach without explicit constraints.
>
> ## Reply to W3: Motivation for Single-Pattern Selection
>
> Thank you for your quesiton.
>
> Our design goal is to teach the model to identify and reinforce the **most effective reasoning strategy** for each specific problem. When different patterns yield different performance levels, we select rollouts from the best-performing pattern. When multiple patterns achieve the same accuracy, we select the pattern that produces the most **concise solution**, thereby jointly optimizing both accuracy and generation efficiency (token cost).
>
> The **"w/o Optimal Pattern Selection"** setting in Table 2 presents experimental results where we update the model using rollouts from **all patterns**, as shown in the table below. We must commend your analysis—it is indeed very reasonable and insightful. Training with rollouts from all patterns achieves better performance on the **GPQA (out-of-domain)** task, but underperforms compared to GPSO on **AIME (in-domain)** tasks. This demonstrates that training with all pattern rollouts may indeed enhance the model's **robustness and generalization**. We sincerely appreciate your valuable insights!
>
> | odel                          | AIME2024 | AIME2025 | GPQA     |
> | ----------------------------- | -------- | -------- | -------- |
> | DeepSeek-R1-Distill-Qwen-7B   | 48.3     | 33.3     | 47.6     |
> | + GPSO                        | **53.3** | **40.0** | 47.9     |
> | w/o Optimal Pattern Selection | 50.8     | 38.3     | **50.3** |
>
> ## Reply to W4:Please Discuss Applicability to Stronger/Larger Models
>
> Thank you for your quesiton.
>
> For stronger and larger models, although they naturally exhibit more diverse reasoning trajectories, they still **do not reliably select the most suitable reasoning pattern for each problem**. As shown in Table 1, even high-performing LRMs such as DeepSeek-R1 and Qwen3-8B (Thinking) can achieve higher ceilings when they are able to choose the optimal pattern for each instance.
>
> As models scale up, the role of GPSO may shift from merely “compensating for insufficient reasoning diversity” to **providing a more effective pattern-selection mechanism**, enabling the model to dynamically switch to the most appropriate reasoning paradigm for different types of problems.

---

> > ### Author Response · Authors · 2025-11-21
> > **Response to Reviewer Hgif**
> >
> > Dear Reviewer,
> > I hope this message finds you well. We have addressed all the concerns you raised. We believe the revisions have significantly strengthened the paper.
> > We would be grateful for any update on the review status at your convenience. Thank you very much for your valuable feedback and time.

---

### Official Review · Reviewer_8397 · 2025-11-04

**Soundness:** 2
**Presentation:** 2
**Contribution:** 3
**Rating:** 4
**Confidence:** 3

**Summary:**

This paper proposes Group Pattern Selection Optimization (GPSO), an RL framework for training reasoning models to adaptively select the right reasoning pattern for a problem. The paper considers three reasoning patterns: direct solution, reflect-and-verify, and explore multiple solutions, each of which is a 1-2 sentence addition to the prompt template (App. B.2). GPSO rolls out multiple responses for each pattern, selects the pattern with the highest empirical accuracy, and updates the policy only on rollouts from the optimal pattern. They mask attention to the suffix tokens during gradient computation to prevent overfitting to explicit prompts. Their experiments show consistent gains across the four reasoning benchmarks.

**Strengths:**

The paper is generally clearly written. It has clear motivation and a good visual presentation.

**Weaknesses:**

- The abstract claims "GPSO learns the intrinsic mapping from problem to pattern", which led me to believe that the model is autonomously discovering reasoning patterns during training. In reality, GPSO selects one of three (or four?) pre-defined prompt templates during training. I think the framing of the paper overstates, and that it's more accurate to view GPSO as an improvement to rollout sampling that prioritizes supervision from high-reward patterns.
- Generally, I think the fact that the method relies on the diversity in a small number of hand-crafted prompt templates is a limiting factor.
- In the intro (line 82-89), you claim that Figure 1 "demonstrates that if LLMs were capable of dynamically selecting the most suitable pattern, their overall performance could be enhanced by a substantial margin". This is a core motivation for your paper. If I understand the Best bar in Figure 1 correctly, you took a question-wise maximum over the different patterns and averaged that value. This sampling procedure is an unfair comparison, similar to pass@k vs pass@1.
- The captions are overall quite sparse and don't contain enough self-contained context

**Questions:**

- Could you comment on computational cost? If I understand correctly, you'd have to roll out three times more examples at each training iteration. If that is correct, perhaps it's reasonable to consider regular GRPO with 3x more iterations as another point of comparison?
- In a few places, you describe your method name as Group Pattern Selection *Optimal* (GPSO). This is a typo, right? I'm assuming you mean Optimization.
- Very minor, but in line 283, you say that your metric is Pass@k with k=1. This is just accuracy; I don't see why you're mentioning Pass@k at all.

---

> ### Author Response · Authors · 2025-11-21
> **Response to Reviewer 8397**
>
> We sincerely thank you for the constructive feedback. We address each concern below and will incorporate these clarifications in our revision.
>
> ## Reply to W1: "Intrinsic mapping" claim overstates the contribution
>
> we acknowledge this is a valid concern about framing. You're right that the current phrasing could mislead readers into thinking the model discovers patterns from scratch. Let me clarify what we actually mean:
>
> - **What we claim**: After GPSO training with attention masking, the model learns to *internally select* which reasoning strategy to employ based on problem characteristics, without requiring explicit pattern prompts at inference time.
>
> The "intrinsic mapping" refers to the model's learned ability to route problems to appropriate pre-defined strategies based on implicit problem features, not pattern discovery. We would revise the abstract and introduction to state more precisely: *"GPSO teaches models to internalize the selection of reasoning patterns."*
>
> ## Reply to W2: Reliance on hand-crafted prompt templates is limiting
>
> Thank you for raising this concern. I want to clarify that our pattern selection is empirically grounded rather than arbitrarily hand-crafted:
>
> We conducted a systematic analysis of 10,000 reasoning trajectories from 7 state-of-the-art LRMs across mathematics and science domains (detailed in Appendix A). Using GPT-4o for classification, we identified that three core patterns—**Direct Solution**, **Reflection and Verification**, and **Explore Multiple Solutions**—account for over 98% of naturally occurring reasoning behaviors in current LRMs. The remaining strategies (Analogy, Reverse Thinking, etc.) appear in fewer than 2% of cases and are consolidated into an "Adaptive" category that allows unconstrained reasoning. Thus, our patterns are not arbitrary design choices but rather a **data-driven characterization** of how existing models already reason.
>
> ## Reply to W3: Best bar in Figure 1 is unfair (like pass@k vs pass@1)
>
> Thank you for this important observation. You're correct that the "Best" bar uses question-wise maximum, but let me clarify our intent:
>
> **Our motivation:** The "Best" bar shows the performance ceiling if a model could select the optimal pattern **on its first attempt** for each problem. This is different from pass@k:
>
> - **Pass@k**: Sample k times with the same strategy, succeed if any attempt works
> - **Our "Best"**: For each question, use whichever single pattern would succeed (oracle pattern selection)
>
> We aim to answer: *"What if the model picks the right reasoning strategy upfront?"* rather than *"What if we try multiple times?"*
>
> **Additional analysis inspired by your question:** Your concern motivated us to investigate: *"How many sampling attempts (pass@k) would be needed to match optimal pattern selection?"* We conducted this experiment using Qwen3-8B (Thinking):
>
> | Model(Qwen3-8B (Thinking))          | AIME2024 | AIME2025 | GPQA     |
> | ----------------------------------- | -------- | -------- | -------- |
> | Pass@1                              | 76.7     | 67.5     | 58       |
> | Best Pattern (Best bar in Figure 1) | **90**   | **80**   | **71.7** |
> | Pass@4                              | 79.2     | 73.3     | 66.2     |
> | Pass@8                              | 83.3     | 76.7     | 71.2     |
>
> Key findings:
>
> - **Pass@8 still underperforms best pattern selection** on AIME2024 (83.3 vs 90.0) and AIME2025 (76.7 vs 80.0)
> - This suggests that **strategic diversity (choosing the right pattern) is more effective than sampling diversity** (multiple attempts with one pattern)
> - Best pattern selection represents a stronger upper bound than typical pass@k baselines
>
> We will add this analysis to the paper and clarify the caption: *"'Best' represents an oracle upper bound under perfect pattern selection on the first attempt.*
>
> ## Reply to W4: Sparse captions lack self-contained context
>
> Thank you for the suggestion, we will expand all captions to be fully self-contained.

---

> > ### Author Response · Authors · 2025-11-21
> > **Response to Reviewer 8397**
> >
> > ## Reply to Q1: Computational cost and comparison to GRPO with 3× rollouts
> >
> > Thank you for this important concern. The baseline you requested is actually already included in **Table 2 as "w/o Multi-Pattern Rollout"**. We appreciate this feedback and will make this comparison more explicit in the revision.
> >
> > In "w/o Multi-Pattern Rollout" ablation setting, we disable pattern-specific rollouts and instead sample the same total number of responses per problem (rollouts = num_patterns × num_samples_per_pattern) without any pattern conditioning. This provides a direct comparison under identical computational budgets.
> >
> > Experimental Setup:
> >
> >  - **GPSO**: 4 patterns × 8 samples = **32 rollouts per problem** (with pattern conditioning), however, only the 8 rollouts of the optimal pattern will be selected for update.
> >  - **w/o Multi-Pattern Rollout**: **32 rollouts per problem** (standard GRPO, no pattern conditioning)
> >
> > The detailed comparison results are shown in the table below.
> >
> > | Model                       | AIME2024 | AIME2025 | GPQA     | Avg.     |
> > | --------------------------- | -------- | -------- | -------- | -------- |
> > | Nemotron-Qwen-1.5B          | 53.3     | 35.8     | 40.5     | 43.2     |
> > | + GRPO                      | 49.2     | 33.3     | 40.5     | 41.0     |
> > | + GPSO                      | **58.3** | **37.5** | **43.2** | **46.3** |
> > | DeepSeek-R1-Distill-Qwen-7B | 48.3     | 33.3     | 47.6     | 43.2     |
> > | + GRPO                      | 50.8     | 35.8     | 47.7     | 44.8     |
> > | + GPSO                      | **53.3** | **40.0** | **47.9** | **47.1** |
> >
> > ## Reply to Q2: "Optimal" vs "Optimization"
> >
> > Thank you for pointing this out. It is a typo, "Optimization" is right, we will correct it in the final version.
> >
> > ## Reply to Q3: Why mention Pass@k when k=1?
> >
> > Thank you! You're right—when k=1, Pass@1 is just accuracy. We will simplify the description.

---

### Note · Authors · 2026-01-06

I have read and agree with the venue's withdrawal policy on behalf of myself and my co-authors.